# Associations of Pain Vigilance and Past and Current Pain with Kinesiophobia after Sport Injury in Current and Former Athletes from Iran and the United States

**DOI:** 10.3390/jfmk8030117

**Published:** 2023-08-12

**Authors:** Fahimeh Badiei, Britton W. Brewer, Judy L. Van Raalte

**Affiliations:** 1Department of Psychology, Springfield College, Springfield, MA 01109, USA; f.badiei77@gmail.com (F.B.); bbrewer@springfieldcollege.edu (B.W.B.); 2College of Health Science, Wuhan Sports University, Wuhan 311112, China

**Keywords:** fear of movement, hypervigilance, memory of pain, reinjury, sport injury

## Abstract

High levels of kinesiophobia (fear of movement/reinjury) have been related to reinjury and adverse injury rehabilitation outcomes in athletes. To examine the extent to which pain vigilance, memory of injury-related pain, and current injury-related pain were associated with kinesiophobia, a cross-sectional study was conducted with 172 current and former athletes from Iran (*n* = 113) and the United States (*n* = 59) who reported having experienced a serious injury that affected their participation or performance in sport. Questionnaires were administered to participants via an online survey platform. Hierarchical multiple regression analysis revealed that pain vigilance and memory of pain were positively associated with kinesiophobia, with the full model accounting for 31% of the variance in kinesiophobia scores. The findings suggest that excessive attention to pain-related stimuli and memory of pain for an injury that occurred an average of four years earlier may contribute to the experience of fear of movement and reinjury in current and former athletes.

## 1. Introduction

Kinesiophobia refers to “an excessive, irrational, and debilitating fear of physical movement and activity resulting from a feeling of vulnerability painful injury or reinjury” [1]. The condition has been widely examined in chronic musculoskeletal pain populations, with strong evidence of a positive association between kinesiophobia and both pain intensity and disability [2]. Kinesiophobia has also been investigated in relation to sport injuries, most extensively with injuries to the anterior cruciate ligament (ACL) [3,4]. In comparison with athletes low in kinesiophobia after injury, those high in kinesiophobia are less likely to return to sport [5] and more likely to incur reinjury [6], report poor physical functioning [7], and exhibit stiffened movement patterns [8]. Athletes with a history of musculoskeletal injury report higher levels of kinesiophobia than those without such a history [9].

Given the ramifications of fear of movement and reinjury for pain, disability, and key sport-injury rehabilitation outcomes, it is important to identify factors associated with postinjury kinesiophobia so that it can be addressed effectively and its adverse impact minimized. Pain intensity is, not surprisingly, a leading predictor of kinesiophobia [2]. Although the majority of research on the association between musculoskeletal pain and kinesiophobia has been cross-sectional and, therefore, focused on current pain, there is compelling reason to believe that perceptions of previously experienced pain may be related to kinesiophobia as well. For athletes who have sustained an injury, the pain associated with that injury may subsequently serve as a reminder of the event that hampered or prevented their involvement in sport. After their injuries have healed, athletes generally return to the environmental and situational context in which the injuries occurred. Memory of injuries that were particularly painful could reasonably be expected to produce apprehension and, potentially, kinesiophobia. In support of this argument, associations have been documented between recalled pain intensity from previous events and levels of pain and distress in subsequent painful experiences [10,11,12]. Further, memory of past pain may contribute to the occurrence of chronic pain [13,14] and decrease willingness to experience situations in which pain stimuli are likely to be encountered [15,16].

Another factor that could contribute to the development of kinesiophobia is pain vigilance, which refers to the tendency to direct attention toward pain-related sensations. People who attend excessively to pain and constantly scan their bodies for somatic and pain sensations are considered to display hypervigilance [17]. Considered an unintentional and efficient process, hypervigilance to pain is closely tied to the experience of fear [18]. Chronic-pain patients [19] and athletes with a history of musculoskeletal injury [9] both tend to attend selectively to pain-related stimuli. Positive associations have been found between pain vigilance and kinesiophobia in multiple samples of chronic-pain patients [20,21]. Consequently, it would be expected that athletes who are hypervigilant to pain would be more likely than those without such a tendency to experience kinesiophobia after injury.

The purpose of the present study was to examine the extent to which pain vigilance, memory of pain from a previous sport injury, and current pain associated with the injury were associated with kinesiophobia among a diverse sample of current and former athletes. Based on previous research on nonathletes with chronic pain [2,20,21], it was hypothesized that pain vigilance, memory of pain, and current pain would be positively associated with kinesiophobia. In the absence of a theoretical rationale for cross-national or cross-cultural differences in contributors to kinesiophobia in current or former athletes, no hypotheses were made with respect to comparisons between participants from Iran with those from the United States in the prediction of kinesiophobia.

## 2. Materials and Methods

### 2.1. Participants

Consistent with research highlighting the importance of conducting sport psychology investigations with diverse groups of participants [22,23,24], samples of current and former athletes were recruited in both Iran and the United States. Participants were 172 individuals (90 men, 81 women, and 1 nonbinary person) from Iran (*n* = 113) and the United States (*n* = 59) with a mean age of 25.24 (*SD* = 7.49) years. Participants from Iran reported current sport involvement at the regional/provincial (*n* = 27), national (*n* = 26), world (*n* = 14), college (*n* = 13), Asian Cup (*n* = 6), and international (*n* = 4) levels (22 participants reported not currently being active in sport). Participants from the United States reported current sport involvement at the intercollegiate (*n* = 19), club (*n* = 6), and intramural (*n* = 2) levels (3 participants did not respond to this item and 29 participants reported not currently being active in sport). Participants reported having sustained an injury that affected their participation or performance in sport an average of 4.17 (*SD* = 4.23) years before involvement in the study, with a majority of participants indicating that their injury required rehabilitation (83%) and pain medication (75%), but not surgery (67%).

### 2.2. Measures

Data pertaining to demographic and injury-related factors, past and current pain, pain vigilance, and kinesiophobia were collected. A questionnaire was used to obtain demographic information (e.g., age, gender, level of past and current sport involvement) and injury-related information (e.g., body location, time since injury occurrence, treatments (i.e., surgery, rehabilitation, and pain medication)). 

Past and current pain were assessed with the short-form McGill Pain Questionnaire (MPQ) [25], which consists of 15 items pertaining to symptoms of various types of pain (i.e., sensory, and affective) that are rated on a scale with response options of 0 (none), 1 (mild), 2 (moderate), and 3 (severe). Support has been documented for the psychometric properties of the short-form MPQ [25] and Persian translations of both the original and short-form versions of the MPQ [26,27]. In the current study, the internal consistency was good for two administrations of the Persian version (Cronbach’s α = 0.88 and 0.95) and the English version (Cronbach’s α = 0.80 and 0.87) of the short-form MPQ.

Pain vigilance was assessed with the Pain Vigilance and Awareness Questionnaire (PVAQ) [28], a 16-item questionnaire on which responses are given on a scale from 0 (never) to 5 (always). Item content pertains to the extent to which respondents attend to the pain that they experience, addressing issues such as awareness, consciousness, observation, and vigilance of pain. Support for the reliability and validity of the PVAQ has been provided [21,28], and a Persian translation of the measure has been used in research with patients with chronic pain [29]. The internal consistency of the PVAQ in the current study was good (Cronbach’s α = 0.88 for the Persian version and 0.85 for the English version).

Kinesiophobia was assessed with the Tampa Scale for Kinesophobia (TSK) [1], which has 17 items pertaining to the fear of movement and reinjury (e.g., “I am afraid that I might injure myself if I exercise”). Responses to TSK items are given on a scale from 1 (strongly disagree) to 4 (strongly agree). Support for the psychometric soundness of the TSK is well documented [30], and the reliability and validity of the Persian version of the scale has been substantiated [31]. The internal consistency of the TSK in the current study was satisfactory (Cronbach’s α = 0.74 for the Persian version and 0.75 for the English version).

### 2.3. Procedure

This study was approved by the Institutional Review Board at Springfield College (#3312122; approved 1 June 2022). Participants were recruited for a study on “the injury experience of people who have been or are currently involved in competitive sport” via email (Iran participants) or an online participant pool for a college introductory psychology course (United States participants). After providing informed consent through a link on the Qualtrics (Provo, Utah) online survey platform at a time and location of their own choosing, participants completed the questionnaire requesting demographic and injury-related information and the PVAQ. Participants who reported that they had not experienced a serious injury that affected their participation or performance in sport were directed out of the survey. Participants who reported that they had experienced a serious injury that affected their participation or performance in sport completed the short-form MPQ with instructions to “select the numbers that best describe the intensity of each of the qualities of pain and related symptoms you felt when your pain associated with the injury was at its worst” (i.e., memory of pain). Participants completed the short-form MPQ again, this time with instructions to “select the numbers that best describe the intensity of each of the qualities of pain and related symptoms you have felt over the past 7 days in the body location of the injury you considered in the previous section of the survey” (current pain). Finally, participants completed the TSK. The order of questionnaire completion corresponded roughly to the temporal sequence in which their potential influence on kinesiophobia was exerted and progressed from demographic, historical, and stable personal variables to more situational variables.

### 2.4. Data Analysis

Data were entered into the IBM Statistical Package for the Social Sciences (SPSS; Armonk, NY, USA) version 24. Data were screened to ensure that only participants who reported experiencing a serious injury that affected their participation or performance in sport were included in the analyses. A *p* < 0.05 significance level was used in all inferential statistical tests.

Descriptive statistics (i.e., means and standard deviations for continuous variables and frequencies for categorical variables) were computed for the main study variables. Cronbach’s alpha coefficients were calculated to assess the internal consistency of the Persian and English versions of each of the multi-item scales used in the study. To identify potential covariates in the prediction of kinesiophobia, Pearson correlations of age and time since injury occurrence with TSK scores were calculated and a series of t-tests comparing TSK scores as a function of gender, sport participation status (i.e., current athlete versus former athlete), and whether participants’ sport injuries required surgery, rehabilitation, or pain medication was conducted. Pearson correlations among the main continuous variables were calculated.

Hierarchical multiple regression analysis was used to predict TSK scores. Stable/dispositional variables (i.e., age, country, and PVAQ scores), MPQ memory of pain scores, MPQ current pain scores, and the six two-way interaction scores involving the main predictor variables (i.e., country, PVAQ, MPQ memory of pain, and MPQ current pain) were entered on the first, second, third, and fourth steps of the analysis, respectively. A one-way multivariate analysis of variance (MANOVA) was performed on age, PVAQ, MPQ memory of pain, and MPQ current pain scores to explicate possible country-related differences in the prediction of TSK scores.

## 3. Results

Means and standard deviations of continuous variables are presented in Table 1. Although time since injury was not significantly correlated with TSK scores (r = −0.01, *p* = 0.90), age was positively associated with TSK scores (r = 0.24, *p* = 0.003) and was, therefore, used as a covariate in the regression analysis. Because the t-tests revealed no significant differences in TSK as a function of gender, sport participation status, or whether participants reported their sport injury as requiring surgery, rehabilitation, or pain medication (all *p* values ≥ 0.10), none of the variables examined in the t-tests were used as covariates in the regression analysis. Intercorrelations among the continuous variables are displayed in Table 2. PVAQ, short-form MPQ (memory of pain), and short-form MPQ (current pain) scores were moderately and positively correlated with TSK scores.

Results of the hierarchical regression analysis predicting TSK scores are displayed in Table 3. In the initial step of the analysis, the block of variables consisting of age, country, and PVAQ scores accounted for a significant percentage of variance in TSK scores. Country and PVAQ scores emerged as significant predictors of TSK scores. Participants from Iran scored higher on the TSK than participants from the United States. PVAQ scores were positively associated with TSK scores. In the second step of the analysis, MPQ (memory of pain) scores contributed significantly to the prediction of (and were positively associated with) TSK scores. In the third step of the analysis, the contribution of MPQ (current pain) scores to the prediction of TSK scores narrowly missed statistical significance (*p* = 0.05). In the final step of the analysis, the block of two-way interaction scores accounted for significant incremental explained variance in TSK scores, but none of the individual interaction terms was statistically significant. The full model regression equation was statistically significant, with R^2^ = 0.31, *F*(11, 132) = 5.41, and *p* < 0.001. 

Results of the one-way MANOVA comparing the age, PVAQ, MPQ memory of pain, and MPQ current pain scores of participants from Iran with those of participants from the United States revealed a significant multivariate effect of country, with Wilks’ lambda = 0.58, *F*(4, 147) = 26.29, *p* < 0.001, and partial eta-squared = 0.42, As shown in Table 4, participants from Iran were significantly older and had significantly higher PVAQ, MPQ memory of pain, and MPQ current pain scores than participants from the United States.

## 4. Discussion

As hypothesized and consistent with research on nonathletes with chronic pain [2,20,21], pain vigilance and pain were both positively associated with kinesiophobia in current and former athletes who reported having sustained an injury that affected their participation or performance in sport. Participants reporting a tendency to attend to pain had higher kinesiophobia scores than those not reporting such a tendency. Although the associations of the two pain variables—memory of pain and current pain—with kinesiophobia were similar in magnitude (both were small effect sizes), only memory of pain was statistically significant. The finding for memory of pain is noteworthy in that the injury to which the recalled pain corresponded was reported to have occurred an average of more than four years earlier. It is, therefore, possible that just as memories of pain from medical procedures [10] and laboratory pain tasks [11,12] are related to the pain experienced in subsequent similar situations, memories of injury-related pain can have a lasting impact on fears of movement and reinjury experienced by current and former athletes. Overall, the predictor variables accounted for 31% of the variance in kinesiophobia, a percentage that may vary as a function of the context and sample characteristics of a given study. 

There are no obvious cross-cultural or cross-national differences between Iran and the United States that account for the association between country and kinesiophobia. It is likely that the higher levels of kinesiophobia reported by participants from Iran relative to those reported by participants from the United States are attributable to other factors that were positively correlated with kinesiophobia in the current study. Compared with participants from the United States, participants from Iran were older and scored higher on pain vigilance, memory of pain, and current pain. Although none of the interactions involving country were statistically significant, the interactions of country with pain vigilance and current pain yielded medium and large effect sizes, respectively, further supporting the explanation that differences in the two countries’ sample characteristics may have underlied the country effect. Additional research on kinesiophobia with athletes around the world will help clarify the generalizability of the current findings. 

It is important to note that in addition to the predictors of kinesiophobia after sport injury identified in this study, multiple factors of potential relevance to the development of fear of movement and reinjury were not associated with kinesiophobia. It would not have been surprising if current and former athletes whose injuries necessitated surgery, rehabilitation, or pain medication had reported higher levels of kinesiophobia than those with less impactful injuries, yet no differences on kinesiophobia were found for these variables. Taken together, significant and nonsignificant findings obtained in the current study are consistent with the fear-avoidance model of pain [32], which holds that pain- and injury-related perceptions are more important than objective pain- and injury-related circumstances (e.g., time since injury occurrence, surgery, rehabilitation, pain medication) in influencing the development of pain- and injury-related fear, excessive attention to pain- and injury-related cues (i.e., hypervigilance), and avoidant pain- and injury-related behavior. Because pain and injury are normalized in sport culture [33], pain-related fear may not necessarily lead to the avoidant behavior in athletes that has been documented for nonathletes in medical and laboratory settings [15,16]. In support of this contention, current athletes and former athletes did not differ on kinesiophobia in the present study. The former athletes can be said to have “escaped” the sport environment, but were no higher in kinesiophobia than the athletes who remained in the sport environment, suggesting that aspects of the fear-avoidance model may not completely generalize to sport participants.

Several limitations of the current study should be considered when interpreting the results. Because a cross-sectional research design was used, it was possible neither to infer causation from the findings nor to establish time-order relationships between the predictor variables and kinesiophobia. To address this issue, experimental and prospective longitudinal studies are recommended for future research on the topic. The current study is also limited by an exclusive reliance on quantitative self-report instruments. Other forms of measurement (e.g., behavioral indicators and otherreports) can be used in future investigations to augment self-report. Qualitative methods may also be useful for identifying ways in which the variables in this study are related and how those relationships have developed over time. In addition, the heterogeneity of the sample should be considered a limitation of the current study. It should be noted, however, that associations between kinesiophobia and potential confounding demographics (i.e., age, gender, and sport-participation status) and injury-related (i.e., time since injury occurrence, surgery, rehabilitation, and pain medication) variables were assessed and, when statistically significant, guided the selection of covariates in the multiple regression analysis.

## 5. Conclusions

Because postinjury kinesiophobia can place athletes at elevated risk for reinjury [6] and other adverse rehabilitation outcomes [5,7,8], it is important to understand more fully the factors associated with its occurrence. In the current study, a model including pain vigilance, memory of pain from a previous sport injury, current pain associated with the injury, and the interactions among these factors accounted for a significant proportion of the variance in kinesiophobia among current and former athletes. Two of the variables—pain vigilance and memory of injury-related pain—were identified as correlates of postinjury kinesiophobia. The tendency to attend to painful bodily sensation and the recollection of a previous sport injury as especially painful were positively associated with kinesiophobia even years after the injury was sustained. Knowledge of such correlates gleaned from prospective, longitudinal, and qualitative studies may inform the development of interventions to address kinesiophobia. Findings from a systematic review and meta-analysis [34] suggest that there is a wide variety of physical (e.g., taping and bracing) and psychological (e.g., mindfulness training and patient education) interventions that can be implemented to reduce kinesiophobia. Further exploration of the efficacy of these interventions and others may help provide practitioners with methods and tools to better serve the athletes with whom they work.

## Figures and Tables

**Table 1 jfmk-08-00117-t001:** Means and standard deviations of continuous variables.

Variable	*M*	*SD*
Age (years)	25.24	7.49
Time since injury (years)	4.17	4.23
PVAQ	46.67	14.35
Short-form MPQ (memory of pain)	46.67	8.92
Short-form MPQ (current pain)	9.08	10.07
TSK	22.03	6.59

*Note*. *n* = 152–169; PVAQ = Pain Vigilance and Awareness Questionnaire; MPQ = McGill Pain Questionnaire; TSK = Tampa Scale for Kinesiophobia.

**Table 2 jfmk-08-00117-t002:** Intercorrelations among continuous variables.

Variable	1	2	3	4	5	6
1 Age (years)	--					
2 Time since injury (years)	0.45 **	--				
3 PVAQ	0.31 **	0.04	--			
4 Short-form MPQ (memory of pain)	0.29 **	0.15	0.28 **	--		
5 Short-form MPQ (current pain)	0.22 *	−0.06	0.28 **	0.53 **	--	
6 TSK	0.24 *	−0.01	0.38 *	0.30 **	0.31 **	--

*Note. n* = 152–169; ** *p* < 0.001; * *p* < 0.01; PVAQ = Pain Vigilance and Awareness Questionnaire; MPQ = McGill Pain Questionnaire; TSK = Tampa Scale for Kinesiophobia.

**Table 3 jfmk-08-00117-t003:** Hierarchical regression analysis predicting TSK scores.

Predictor	ΔR^2^	R^2^	β	*F*	*F* _Change_
Step 1	0.20	0.20		11.29 **	11.29 **
Age			0.07		
Country			0.19 *		
PVAQ			0.28 **		
Step 2	0.03	0.22		9.76 **	4.37 *
MPQ (memory)			0.17 *		
Step 3	0.02	0.24		8.74 **	3.68 *
MPQ (current)			0.18		
Step 4	0.07	0.31		5.41 **	2.23 *
Country × PVAQ			0.41		
Country × MPQ (memory)			−0.18		
Country × MPQ (current)			−0.50		
PVAQ × MPQ (memory)			0.12		
PVAQ × MPQ (current)			−0.04		
MPQ (memory) ×			−0.54		
MPQ (current)					

*Note. n* = 144. PVAQ = Pain Vigilance and Awareness Questionnaire; MPQ = short-form McGill Pain Questionnaire; TSK = Tampa Scale for Kinesiophobia. ** *p* ≤ 0.001; * *p* ≤ 0.05.

**Table 4 jfmk-08-00117-t004:** Means and standard deviations of continuous predictor variables by country.

Variable	Iran		United States	
	*M*	*SD*	*M*	*SD*
Age (years) *	28.13	7.61	19.78	2.17
PVAQ *	50.45	13.90	38.90	12.97
Short-form MPQ (memory of pain) *	24.53	9.46	18.80	7.21
Short-form MPQ (current pain) *	12.13	10.94	4.00	5.44

*Note*. *n* = 152; PVAQ = Pain Vigilance and Awareness Questionnaire; MPQ = McGill Pain Questionnaire; TSK = Tampa Scale for Kinesiophobia. * Country difference significant at *p* < 0.001.

## Data Availability

The data used in this study are available from the corresponding author upon reasonable request.

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
