# Peer review of "Associations of Pain Vigilance and Past and Current Pain with Kinesiophobia after Sport Injury in Current and Former Athletes from Iran and the United States"

_jfmk, 2023, doi:10.3390/jfmk8030117_

Round 1
Reviewer 1 Report
In the following, I will present some comments on the different sections of the paper.
First, I would like to point out the real interest of the object of study of this paper. Although it is a work with several limitations, as the authors point out, I believe that it is a work that can describe a reality that can guide practical applications of various kinds in the future. However, I believe there are some issues that need to be considered for implementation.
1. In the introduction, attention should be paid to including more content. More information on pain vigilance and past and current pain is needed. It would be advisable to include more references.
2. Similarly, the discussion section should be expanded along the same lines. It would be advisable to include more references.
3. More information from participants is needed. It is important to better categorise the sample. The authors establish relationships between different groups of athletes (current and former athletes, i.e.). Therefore, I suggest that this information is included in the sample section.
4. I have a question in the measures section. Are the questionnaires used organised according to factors? The authors explain that the instruments have a certain number of items, but it is not known whether these items refer to the same factor. This is an important aspect, because if factors are present in each questionnaire, it is necessary to include the internal consistency analysis by factors as well.
5. In "procedure" section, it would be interesting to include the following information:
- How did the participants complete the questionnaires? Was it online? Was it at a specific location? Timeliness? Information on this is needed in order to consider the validity of the data.
- The authors comment on the order in which the questionnaires were completed. I consider it necessary to include information justifying this order, for the reader's information.
6. The "conclusions" section needs to be improved. The authors should elaborate further on their findings. As a suggestion, I think it would be interesting to include practical applications to answer the question: what can be done from here?
7. Expand the references it is necessary.
Congratulations for the paper and effort.
Author Response
Reviewer 1
In the following, I will present some comments on the different sections of the paper.
First, I would like to point out the real interest of the object of study of this paper. Although it is a work with several limitations, as the authors point out, I believe that it is a work that can describe a reality that can guide practical applications of various kinds in the future. However, I believe there are some issues that need to be considered for implementation.
Thank you for your time and effort in reading and providing feedback on the manuscript. Your feedback has been invaluable in preparing the revision.
- In the introduction, attention should be paid to including more content. More information on pain vigilance and past and current pain is needed. It would be advisable to include more references.
Additional information and references on pain vigilance and memory of pain have been provided in new material on lines 47-51 and 54-57 as follows:
“In support of this argument, associations have been documented between recalled pain intensity from previous events and levels of pain and distress in subsequent painful experiences [10-12]. Further, memory of past pain may contribute to the occurrence of chronic pain [13, 14] and decrease willingness to experience situations in which pain stimuli are likely to be encountered [15,16].”
“People who attend excessively to pain and constantly scan their bodies for somatic and pain sensations are considered to display hypervigilance [17]. Considered an unintentional and efficient process, hypervigilance to pain is closely tied to the experience of fear [18].”
- Similarly, the discussion section should be expanded along the same lines. It would be advisable to include more references.
We have linked the findings of the study to new material presented in the Introduction by adding 5 references to the Discussion section on lines 274-277 and 304-306.
“It is, therefore, possible that just as memories of pain from medical procedures [10] and laboratory pain tasks [11,12] are related to the pain experience in subsequent similar situations, memories of injury-related pain can have a lasting impact on fears of movement and reinjury experienced by current and former athletes.”
“Because pain and injury are normalized in sport culture [33], however, pain-related fear may not necessarily lead to the avoidant behavior in athletes that has been documented for nonathletes in medical and laboratory settings [15,16].”
- More information from participants is needed. It is important to better categorise the sample. The authors establish relationships between different groups of athletes (current and former athletes, i.e.). Therefore, I suggest that this information is included in the sample section.
In addition to the information on the number of participants who were former athletes provided on lines 80, 81, 83, and 84, more injury-related information about participants is given in new material on lines 84-87 as follows:
“Participants reported having sustained an injury that affected their participation or performance in sport an average of 4.17 (SD = 4.23) years before involvement in the study, with majorities of participants indicating that their injury required rehabilitation (83%) and pain medication (75%), but not surgery (67%).”
- I have a question in the measures section. Are the questionnaires used organised according to factors? The authors explain that the instruments have a certain number of items, but it is not known whether these items refer to the same factor. This is an important aspect, because if factors are present in each questionnaire, it is necessary to include the internal consistency analysis by factors as well.
Because total scores were used for all three of the standardized questionnaires, a single internal consistency value was obtained and reported for each questionnaire.
- In "procedure" section, it would be interesting to include the following information:
- How did the participants complete the questionnaires? Was it online? Was it at a specific location? Timeliness? Information on this is needed in order to consider the validity of the data.
As noted in material on lines 125-126, the data were collected online at a time and location of participants’ own choosing.
- The authors comment on the order in which the questionnaires were completed. I consider it necessary to include information justifying this order, for the reader's information.
As noted in new material on lines 138-141, “The order of questionnaire completion corresponded roughly to the temporal sequence in which their potential influence on kinesiophobia was exerted and progressed from demographic, historical, and stable personal variables to more situational variables.”
- The "conclusions" section needs to be improved. The authors should elaborate further on their findings. As a suggestion, I think it would be interesting to include practical applications to answer the question: what can be done from here?
We have elaborated on the findings and expanded the Conclusions section in new material on lines 330-333, 335-337, and 339-344 as follows:
“In the current study, a model including pain vigilance, memory of pain from a previous sport injury, current pain associated with the injury, and the interactions among these factors accounted for a significant proportion of the variance in kinesiophobia among current and former athletes.”
“The tendency to attend to painful bodily sensation and the recollection of a previous sport injury as especially painful were positively associated with kinesiophobia even years after the injury was sustained.”
“Findings from a systematic review and meta-analysis [34] suggest that there is a wide variety of physical (e.g., taping, bracing) and psychological (e.g., mindfulness training, patient education) interventions that can be implemented to reduce kinesiophobia. Further exploration of the efficacy of these interventions and others may help provide practitioners with methods and tools to better serve the athletes with whom they work.”
- Expand the references it is necessary.
In the course of revising the manuscript, the list of references has been expanded. A total of 10 references has been added, thereby increasing the number of references by 42%.
Congratulations for the paper and effort.
Thank you for your kind words and helpful comments.
Reviewer 2 Report
First of all, I appreciate the opportunity to review the paper. I would like to provide a series of recommendations to the authors:
Lines 57-67: All references to the method should be included in the "Materials and Methods" section. For example, "Consistent with research highlighting the importance of conducting sport psychology investigations with diverse groups of participants [13-15], samples of current and former athletes were recruited in both Iran and the United States." References to the method should be placed in the method section.
One issue I see with the research is that the sample is highly diverse, which introduces biases in the results. Therefore, I suggest that the authors include this aspect as a limitation of the study in the Discussion section.
Lines 130-153: In my opinion, it is not necessary to divide the Data analysis into two parts, "Preliminary Analyses" and "Main Analyses." Therefore, my recommendation is to unify them. Additionally, in this section, the p-value for the significance level should be indicated in the statistical estimates.
Regarding the results in Table 1 (Lines 209-230), the overall final hierarchical regression model with all variables included in Steps 1, 2, 3, and 4 explained a total of 31% of the variance in TSK scores. This means that the variables introduced in each step gradually improved the model's ability to predict TSK scores. However, it is important to note that the level of prediction may vary depending on the context and specific characteristics of this study. This analysis should be better reflected in the discussion.
Lines 297-305: The limitation of having a heterogeneous sample should be included.
I recommend that the authors review the conclusions. The conclusions should address the objective, which is not clear in this case. Therefore, I ask the authors to revise this matter.
In the objective, they state: "The purpose of the present study was to examine the extent to which pain vigilance, memory of pain from a previous sport injury, and current pain associated with the injury were associated with kinesiophobia among current and former athletes. Consistent with research highlighting the importance of conducting sport psychology investigations with diverse groups of participants..."
However, in the conclusion, they state: "Because post-injury kinesiophobia can increase the risk of reinjury in athletes and lead to other adverse outcomes in rehabilitation, it is important to further understand the factors associated with its occurrence. In the current study, two variables, pain vigilance and memory of injury-related pain, were identified as correlated with post-injury kinesiophobia. Knowledge of such correlations can aid in the development of interventions to address kinesiophobia and enable professionals to better assist the athletes they work with."
As can be seen, the discussion does not provide an answer to what was proposed in the objective.
Author Response
Reviewer 2
First of all, I appreciate the opportunity to review the paper. I would like to provide a series of recommendations to the authors:
Thank you for your time and effort in reading and providing feedback on the manuscript. Your feedback has been invaluable in preparing the revision.
Lines 57-67: All references to the method should be included in the "Materials and Methods" section. For example, "Consistent with research highlighting the importance of conducting sport psychology investigations with diverse groups of participants [13-15], samples of current and former athletes were recruited in both Iran and the United States." References to the method should be placed in the method section.
As suggested, the sentence in question has been moved to the Materials and Methods section (lines 74-76).
One issue I see with the research is that the sample is highly diverse, which introduces biases in the results. Therefore, I suggest that the authors include this aspect as a limitation of the study in the Discussion section.
We have included the heterogeneity of the sample as a limitation of the study in the Discussion section on lines 321-326 as follows:
“In addition, the heterogeneity of the sample should be considered a limitation of the current study. It should be noted, however, that associations between kinesiophobia and potential confounding demographic (i.e., age, gender, sport participation status) and injury-related (i.e., time since injury occurrence, surgery, rehabilitation, pain medication) variables were assessed and, when statistically significant, guided the selection of covariates in the multiple regression analysis.”
Lines 130-153: In my opinion, it is not necessary to divide the Data analysis into two parts, "Preliminary Analyses" and "Main Analyses." Therefore, my recommendation is to unify them. Additionally, in this section, the p-value for the significance level should be indicated in the statistical estimates.
As recommended, we have unified the preliminary analyses and the main analyses in the Data Analysis and Results sections. The significance level for all inferential statistical tests has been noted in new material on lines 146-147 as follows:
“A p < .05 significance level was used in all inferential statistical tests.”
Regarding the results in Table 1 (Lines 209-230), the overall final hierarchical regression model with all variables included in Steps 1, 2, 3, and 4 explained a total of 31% of the variance in TSK scores. This means that the variables introduced in each step gradually improved the model's ability to predict TSK scores. However, it is important to note that the level of prediction may vary depending on the context and specific characteristics of this study. This analysis should be better reflected in the discussion.
The fact that the percentage of variance in kinesiophobia may vary as a function of the context and sample characteristics of a given study is noted in new material on lines 277-279 as follows:
“Overall, the predictor variables accounted for 31% of the variance in kinesiophobia, a percentage that may vary as a function of the context and sample characteristics of a given study.”
Lines 297-305: The limitation of having a heterogeneous sample should be included.
We have acknowledged the heterogeneity of the sample as a limitation of the study in new material on lines 321-326 as follows:
“In addition, the heterogeneity of the sample should be considered a limitation of the current study. It should be noted, however, that associations between kinesiophobia and potential confounding demographic (i.e., age, gender, sport participation status) and injury-related (i.e., time since injury occurrence, surgery, rehabilitation, pain medication) variables were assessed and, when statistically significant, guided the selection of covariates in the multiple regression analysis.”
I recommend that the authors review the conclusions. The conclusions should address the objective, which is not clear in this case. Therefore, I ask the authors to revise this matter.
In the objective, they state: "The purpose of the present study was to examine the extent to which pain vigilance, memory of pain from a previous sport injury, and current pain associated with the injury were associated with kinesiophobia among current and former athletes. Consistent with research highlighting the importance of conducting sport psychology investigations with diverse groups of participants..."
However, in the conclusion, they state: "Because post-injury kinesiophobia can increase the risk of reinjury in athletes and lead to other adverse outcomes in rehabilitation, it is important to further understand the factors associated with its occurrence. In the current study, two variables, pain vigilance and memory of injury-related pain, were identified as correlated with post-injury kinesiophobia. Knowledge of such correlations can aid in the development of interventions to address kinesiophobia and enable professionals to better assist the athletes they work with."
As can be seen, the discussion does not provide an answer to what was proposed in the objective.
The following sentence has been added to the Conclusion on lines 330-333 to better align the content of the section with the stated purpose of the study:
“In the current study, a model including pain vigilance, memory of pain from a previous sport injury, current pain associated with the injury, and the interactions among these factors accounted for a significant proportion of the variance in kinesiophobia among current and former athletes.”